# Whey Protein–Tannic Acid Conjugate Stabilized Emulsion-Type Pork Sausages: A Focus on Lipid Oxidation and Physicochemical Features

**DOI:** 10.3390/foods12142766

**Published:** 2023-07-20

**Authors:** Tanong Aewsiri, Palanivel Ganesan, Hataikan Thongzai

**Affiliations:** 1School of Agricultural Technology and Food Industry, Walailak University, Nakhon Si Thammarat 80160, Thailand; kanhataikanthongzai@gmail.com; 2Department of Biotechnology, College of Biomedical and Health Science, Nanotechnology Research Center, Konkuk University, Chungju 27478, Republic of Korea; palanivel67@gmail.com

**Keywords:** phenolic-protein conjugate, protein modification, functional properties, meat, stability

## Abstract

The purpose of this study was to investigate the oxidative stability and physicochemical properties of pork emulsion sausages with whey protein–tannic acid conjugate and native whey protein. Over the course of 21 days, the thiobarbituric acid reactive substances (TBARS) of sausages containing a whey protein–tannic acid conjugate were lower than those of sausages with regular whey protein (*p* < 0.05). Kinetically, sausage containing the whey protein–tannic acid conjugate (k = 0.0242 day^−1^) appeared to last longer than sausage containing regular whey protein (k = 0.0667 day^−1^). The addition of the whey protein–tannic acid conjugate had no effect on product texture because there was no difference in hardness, springiness, cohesiveness, or water-holding capacity between the control and treated samples at Day 0 (*p* > 0.05). Scanning electron microscopy revealed that, at Day 21, the control sausage exhibited emulsion coalescence, as evidenced by an increase in the number of oil droplets and large voids, but not the whey protein–tannic acid conjugate-added sausage. There was no variation in the *L**, *a**, and *b** values of the sausages when the whey protein–tannic acid conjugate was added (*p* > 0.05). However, there was a little increase in ΔE value in the treated sample. Thus, the whey-protein–tannic acid conjugate appeared to stabilize the lipid and physicochemical properties of the sausages by lowering the rate of TBARS production, retaining texture, water-holding capacity, and color, as well as by minimizing lipid coalescence during refrigerated storage.

## 1. Introduction

For more than a hundred years, emulsion-type meat products that are commonly consumed in most countries have been prepared using emulsification technology [1]. Because they are convenient and may be used in a variety of cuisines, it has been reported that emulsion-type sausages are more widely consumed than other processed meat products [2]. Emulsion stability between the fat and water phases is one of the most crucial quality traits for processed meat products like emulsified sausages [3]. Additionally, an increasing variety of ingredients and additives are used in the production of recent meat products to enhance various aspects of the product, such as flavor, techno-functional qualities, and storage stability [4]. Soy, casein, whey, egg white, gelatin, protein hydrolysates, and blood plasma are the proteins most frequently found in protein preparations used as emulsifiers and stabilizers in the creation of emulsion-type sausages [4,5]. The key factors influencing their extensive use are their significant capacity to emulsify fat, stabilize the emulsion, and have substantial water-binding properties [4]. These proteins can be employed to lower production and product costs while improving the product’s texture and cooking efficiency. Due to their nutritional and healthful properties or a lower energy value than animal fat, vegetable or whey preparations utilized in higher quantities become good replacements for meat and fat components [4].

The primary protein in milk, whey protein, contains several globular proteins that are well known for acting as natural emulsifiers in food products [6]. However, whey protein has been found to have poor emulsifying stability [7]. To bypass these restrictions, protein modification or binding with other molecules (such as polysaccharides, polyphenols, or proteins, etc.) is frequently performed [8]. Covalent bonds do not easily break under some extreme conditions, such as heating, acidic or alkali conditions, and so on [9,10], making covalent-binding interaction or molecular modification a good solution to this issue. Molecular modification is an intriguing technique for creating novel materials with desired chemical and physical characteristics [11]. Because of polyphenols’ numerous functions, protein–polyphenol conjugations have garnered a lot of interest. It has been observed that polyphenols can react with proteins under oxidizing circumstances, resulting in the creation of protein–polyphenol conjugates [10]. Covalent crosslinking reactions between proteins and polyphenols can improve protein functional capabilities such as gelling, foaming, and emulsifying qualities [12,13].

Tannic acid is a polyphenol with favorable physiological effects (such as antioxidant and antibacterial properties), and it can be utilized to create composite films as sustainable antioxidant and barrier packaging materials. Tannic acid can be modified to improve its antibacterial activity and biocompatibility for biomedical purposes [14,15]. Tannic acid at a level of 200 ppm was found to be effective in retarding lipid oxidation and off-odor development in ground and cooked fish (*Scomberomorus commersoni*) stored at 4 °C [16]. It has been reported that tannic acid at a concentration of 200 ppm inhibited lipid oxidation and the occurrence of off-odor in striped catfish slices and ground beef wrapped in modified atmospheric packaging during refrigeration [17,18]. Protein -NH_2_ and -SH residues were able to form covalent C-N or C-S bonds with the C=O or C-H groups of tannic acid under alkaline circumstances, resulting in protein–tannic acid conjugates [19]. It was discovered that the conjugate-stabilized emulsion was more stable against heat treatment and centrifugation compared to unmodified whey protein [7]. Covalent crosslinking in whey protein–tannic acid conjugates generated a network based on tannic acid molecules acting as bridges between emulsion droplets, which was significant in the stabilizing impact [7,19]. Covalent connections were formed between whey protein and tannic acid, according to the results. Whey protein–tannic acid conjugates outperformed free whey protein in terms of oil-water interfacial activity and wettability [7,20].

One of the significant processes involved in the deterioration of emulsified sausages and other meat products is lipid oxidation. The most frequent way for lipid oxidation to happen is by oxidative radical reactions. Higher numbers of double bonds within the fatty acid enhance the likelihood of autoxidation, which happens when an oxygen ion substitutes a hydrogen ion within a lipid molecule [21,22]. Additionally, pro-oxidants such as metal ions, elevated temperature, light, and the presence of oxygen all affect lipid oxidation [22]. The technology used in meat processing allows for the delay of lipid oxidation. Numerous investigations have generally been conducted on synthetic or natural antioxidants. The lipid oxidation of meat products can be significantly inhibited by synthetic antioxidants such as butylated hydroxyanisole (BHA), butylated hydroxytoluene (BHT), and tertiary butyl hydroquinone (TBHQ) [23,24]. Consumer worries about synthetic antioxidants have, however, grown dramatically [25]. As a result, there has been a steady rise in interest in raw materials that include bioactive or functional components [26]. The manufacture of healthier meat products with a natural antioxidant is now being explored using natural additives derived from nuts, fruits, vegetables, herbs, and spices [27,28,29,30]. The action of phenolic compounds is primarily responsible for the antioxidant effect of such natural additives [27].

Furthermore, the protein–phenolic conjugate has been shown to be a natural antioxidant for meat products [31]. Whey protein is a typical food addition used to improve product stability and texture, whereas phenolics are food antioxidants [20]. As a result, coupling whey protein with phenolics is an efficient strategy to improve protein functioning while preserving polyphenol bioactivity [20]. In a prior work [20], whey protein coupled with 5% oxidized tannic acid displayed the highest antioxidative activity, as determined by 1,1-diphenyl-2-picrylhydrazyl (DPPH), 2,2-azinobiz-ethylbenzothiozolie-6-sulphonic acid (ABTS), and ferric reducing antioxidant power (FRAP) assays, along with an enhanced emulsion stability index in vitro. The linkage between whey protein and phenolic compounds was validated by a loss in free amino content and an increase in total phenolic content of the modified whey protein after it was treated with oxidized phenolic compounds [20]. Tannic acid with a high hydroxyl and carboxyl group concentration was incorporated into whey protein, which increased its hydrophilicity. When the hydrophilic group increased faster than the hydrophobic group, the surface hydrophobicity of the whey protein treated with oxidized tannic acid decreased [20]. It was proposed that whey protein treated with 5% oxidized tannic acid might be employed as an antioxidative food additive in emulsified food products.

As a result, the purpose of this study was to investigate the impact of the whey protein–tannic acid conjugate on the oxidative stability and physicochemical features of pork emulsion sausages in comparison with unmodified whey protein. The findings can be utilized to provide guidelines for the usage of whey protein–tannic acid as an effective ingredient capable of both antioxidant and emulsifying activities in emulsified meat products. This ingredient may help to maintain oxidative and emulsion stability, textural properties, and color of the products, resulting in quality enhancement that benefits meat product manufacturers.

## 2. Materials and Methods

### 2.1. Chemicals

Whey protein concentrate (WPC) (82.1% protein) was obtained from I.P.S. International Co., Ltd. (Bangkok, Thailand). Tannic acid, as well as all chemicals such as Folin-Ciocalteu’s phenol reagent, sodium hydroxide (NaOH), hydrochloric acid (HCl), trichloroacetic acid (TCA), and 2-thiobarbituric acid (TBA), were acquired from Sigma-Aldrich Corp. (St. Louis, MO, USA).

### 2.2. Preparation of Whey Protein–Tannic Acid Conjugate

The method of Aewsiri et al. [20] was used to prepare a whey protein–tannic acid conjugate. Tannic acid is required to oxidize to quinone at a high pH range–pH 9 in this case–to allow the covalent interaction between tannic acid and whey protein. Briefly, tannic acid was dissolved in distilled water at 5% *w*/*w* concentration, then pH was adjusted to 9 with 1 M NaOH. To convert phenolic compounds into an oxidized state, solutions were constantly agitated at room temperature (27–29 °C) for an hour with free exposure to air. A 75 mL whey protein solution (2% protein, *w*/*v*) was combined with oxidized tannic acid solution at 5.0% based on protein content. To obtain a final concentration of 1.5% protein, the final volume was adjusted to 90 mL using distilled water. For a period of 3 h, the solutions were continuously mixed at room temperature. Following that, the solutions were dialyzed at room temperature (molecular weight cut-off = 14,000 Da) for 24 h against 20 vol of distilled water to eliminate residual phenolic compounds. The pH of the conjugate after dialysis was around 7.0. All samples were freeze-dried and stored at −20 °C. The decrease in free animo group content and surface hydrophobicity was used in monitoring the formation of the whey protein–tannic acid conjugation [20]. The free amino group was reduced from 565 to 367 μmol *L*-leucine/100 g sample (about 35% reduction), while the surface hydrophobicity (S_0_ of 1-anilinonaphthalene-8-sulphonic acid; S_0_ANS) was reduced from 1173 to 110 (a reduction of about 91%). Figure 1 depicts the proposed mechanism for covalent conjugation of protein and polyphenol.

### 2.3. Preparation of Pork Emulsion Sausages

Fresh pork loin (*Longissimus dorsi*) was acquired in Thasala’s market in Nakhon Si Thamarat, Thailand. During transportation to Walailak University’s Department of Food Technology, the pork was stored on ice. When the pork arrived, it was rinsed with tap water, trimmed, and minced with a mincer. To maintain consistency between lots, the moisture content of the mince was set at 86%. The pork mince (86.25 g) was combined with sodium chloride (2 g), sodium tripolyphosphate (0.25 g), whey protein concentrate (control) or whey protein–tannic acid conjugate (1.5 g), and soy bean oil (10 g) to make pork emulsion sausage. In contrast to pork backfat, which mostly contains saturated fat, soybean oil was intended to be used in this study since it can be simpler to oxidize and generate coalescence. With this oil, it may be easier for the system to keep track of emulsion instability and lipid oxidation. To emphasize the influence of the whey protein–tannic acid conjugate on lipid oxidation and physicochemical qualities, spices and sodium nitrite were left out of the formula. The mixture was ground for 5 min in a Panasonic food blender (MK, 5087 M, Semenyih, Selangor, Malaysia) to obtain a homogenous paste. The paste was put into a cellophane casing (diameter 20 mm) and cooked in a temperature-regulated water bath (Memmert, D-91126, Schwabach, Germany) for 15 min at 80 °C. To maintain food safety, the product’s core temperature was no lower than 72 °C [27]. The samples were then submerged in ice water for about 30 min before being packed individually in polyethylene bags and refrigerated for 21 days (4–5 °C). The pH values of all samples ranged from 6.7 to 6.8, and there was no significant difference over the course of storage. For analysis, samples were obtained at random on Days 0, 7, 14, and 21 of storage.

### 2.4. Physio-Chemical Properties

#### 2.4.1. Thiobarbituric Acid Reactive Substances (TBARS) and Kinetic Study

TBARS were investigated on Days 0, 7, 14, and 21 using the method outlined by Buege and Aust [32]. A standard curve was created using 1,1,3,3-tetramethoxypropane at concentrations ranging from 0 to 10 ppm. TBARS were represented as mg of malondialdehyde (MDA) equivalent/kg sample.

To study the kinetic changes in TBARS during storage, the first-order kinetic model was used [33].
(1)$$ln\left[\frac{{TBARS}_{t}}{{TBARS}_{0}}\right]=kt$$

TBARS_t_ and TBARS_0_ indicated the TBARS contents at time t and time 0, respectively. The slope of the graph was used to calculate the reaction rate constant (k).

#### 2.4.2. Texture and Water-Holding Capacity

Hardness (N), springiness (mm), and cohesiveness of samples were measured using the methods described by Maqsood et al. [34], using a texture analyzer (LR 5K; LLOYD instruments, West Sussex, UK). Before analysis, the samples were left at room temperature for 1 h to equilibrate before being cut into cylinders (2 × 2 cm^2^). With a cylindrical aluminum probe with a diameter of 70 mm, two cycles of compression were carried out at a test speed of 1.0 mm/s (or 50% of the initial gel height) and trigger force of 0.05 N.

For water-holding capacity, the sausage sample (20 mm diameter/5 mm thickness) was weighed (W1). It was then sandwiched between two layers of Whatman paper No. 1 (2 on top and 3 on the bottom). A 1 kg reference mass was placed on the sample and held for 2 min. The compressed sample was then weighed again (W2). The water capacity was calculated and given as a percentage of the sample weight [35].
(2)$$water\;holding\;capacity\left(\%\right)=\left[\frac{W2}{W1}\right]\times 100$$

#### 2.4.3. Microstructure

A scanning electron microscope (SEM) (GeminiSEM; Carl Ziess Microscopy, Oberkochem, Germany) was used to examine the microstructures of cooked pork emulsion sausages [36]. The samples were cut into 1–2 mm cubes, fixed with 2.5% (*v*/*v*) glutaraldehyde in 0.2 M phosphate buffer (pH 7.2), and then rinsed twice with 0.1 M phosphate buffer and distilled water for 2 h at room temperature. After that, ethanol was used to dehydrate the fixed samples for 10 min at various concentrations of 50%, 70%, 80%, 90%, and 100%. After critical-point drying with CO_2_ as the transition fluid, the dried samples were mounted on a bronze stub and given a gold sputter coating. The samples were scanned using an SEM with a 3 kV acceleration voltage and a magnification of 200×.

#### 2.4.4. Color

The color of the pork emulsion sausages was measured using a colorimeter Hunterlab Miniscan/EX instrument (10° standard observers, illuminant D65; Hunter Assoc. Laboratory, Reston, VA, USA). The color values were reported in the CIE color profile system as *L** value (lightness), *a** value (redness/greenness), and *b** value (yellowness/blueness). Total differences in color (ΔE) between the control sausage sample at Day 0 and that of other samples at Days 0, 7, 14, and 21 were calculated [35].
(3)$$\Delta E=\sqrt{{\left(\Delta L^{*}\right)}^{2}+{\left(\Delta a^{*}\right)}^{2}+{\left(\Delta b^{*}\right)}^{2}}$$
where Δ*L**, Δ*a**, and Δ*b** are the differences between the color parameters of the samples and those of the control at Day 0.

### 2.5. Statistical Analysis

All experiments were carried out in triplicate (*n* = 3) and the data were reported by mean ± standard deviation. An ANOVA analysis was performed on the data. Duncan’s multiple range test was used to compare the means. To compare two variables, the *t*-test was used. For the statistical analysis, SPSS 23.0 (SPSS Inc., Chicago, IL, USA) was utilized.

## 3. Results and Discussion

### 3.1. Lipid Oxidation and Kinetic Study

Figure 2a depicts the effect of the whey protein–tannic acid conjugate on lipid oxidation, as evaluated by TBARS of pork emulsion sausage, during refrigerated storage in comparison to a control formula containing native whey protein. At day 0, the TBARS values of sausages with the whey protein–tannic acid conjugate and control were 1.14 and 1.45 mg MDA equivalent/kg of sample, respectively, showing that lipid oxidation occurred during the sausage’s manufacturing and cooking. Based on the results, the TBARS of sausages with the added whey protein–tannic acid conjugate were lower than those with added typical whey protein throughout the course of 21 days. The prevention of lipid oxidation of the whey protein–tannic acid conjugate can be seen clearly on Days 14 and 21 of storage, implying the potential to increase the shelf-life of sausages by maintaining the oxidative stability of the products. In general, lipids can be classified as not oxidized (TBARS value < 1.5 mg MDA/kg), moderately oxidized (1.6 < TBARS value < 3.6), or oxidized (TBARS value > 3.7), according to Larouche et al. [37]. According to the permissible requirement for the TBARS level, the control sample can be stored for 7 days; however, the sample containing the whey protein–tannic acid conjugate can be stored for up to 21 days. The tannic acid moiety inserted in the whey protein molecule may be responsible for the antioxidant action [17,18,20]. Tannic acid demonstrated radical scavenging ability through hydrogen donation and reducing power, hence halting propagation [17,18]. Tannic acid, in addition to serving as a radical scavenger, has the potential to chelate iron, which may be produced during cooking. Tannic acid can chelate iron, especially in its free form [38,39]. Tannic acid chelates iron due to its ten galloyl groups and, like other iron chelators, may be able to block iron-mediated oxyradical production [38,39]. Tannic acid was recently shown to have stronger ferric-reducing antioxidant power (FRAP) than other phenolic compounds (catechin, ferulic acid, and caffeic acid) [40]. Tannic acid (0.02% and 0.04%) effectively delayed lipid oxidation of fish emulsion sausages during 20 days of refrigerated storage [17,18]. The findings agree with Wongnen et al. [27], who reported the antioxidant acidity of a phenolic extract from *Glochidion wallichianum* leaf as a natural antioxidant in a pork sausage model system. Quan et al. [31] demonstrated that the interaction between proteins and polyphenols is critical in boosting the quality of various foods. As a result, protein–polyphenol conjugates have been proposed as effective antioxidant emulsifiers capable of locating and acting at the oil/water interface, hence minimizing oxidation in emulsified foods [31]. During the manufacturing of emulsified meat products, heating also induces protein interactions between the continuous phase and the membrane enclosing the surface of the oil droplets, limiting lipid coalescence/water flow, and resulting in a stable product [41].

A kinetic analysis of the increase in lipid oxidation evaluated by TBARS of pork sausages during storage was carried out (Figure 2b). Changes in the oxidative indicator, TBARS, among samples over time corresponded to a first-order kinetic scenario (R^2^ = 0.8686–0.9666). According to the findings, lipid oxidation of sausages occurs continuously during storage. Based on the results, sausages containing the whey protein–tannic acid conjugate had a lower growing rate of TBARS than the control (Figure 2b). Sausage containing the whey protein–tannic acid conjugate (k = 0.0242 day^−1^) appeared to persist longer than those containing typical whey protein (k = 0.0667 day^−1^). As a consequence, the whey protein–tannic acid conjugate was effective in reducing lipid oxidation in pork emulsion sausage during refrigerated storage to roughly three times slower.

### 3.2. Texture and Water-Holding Capacity

It has been shown that employing phenolic compounds improved emulsification and gel formation of myofibrillar proteins and also increased the oxidative stability of meat emulsions without affecting textural features [42]. The optimal amount of oxidized phenolic compounds, depending on the particular type of phenolic, could improve the textural qualities of bigeye snapper surimi [43]. The covalent whey protein–tannic acid compound produced in an alkaline environment was found to have higher interfacial activity and wettability than free whey protein [7]. In this investigation, the addition of the whey protein–tannic acid conjugate had no effect on product hardness because there was no difference in hardness between the control and treated samples at the beginning (Day 0) (*p* > 0.05; Table 1). According to Hayes et al. [44], the addition of phenolic components such as lutein, ellagic acid, and sesamol had no effect on the overall texture, tenderness, and flavor of cooked pork sausage stored at 4 °C. Herein, hardness was considerably reduced for control on Day 7 and steadily declined until the end of storage. The whey protein–tannic acid conjugate, on the other hand, appeared to stabilize hardness of sausages more efficiently throughout storage, as seen by stable hardness up to Day 7, then gradually dropping at Day 14 and remaining constant until the end of storage. From Day 7 to Day 21, the treated samples had higher hardness values than controls, indicating the whey protein–tannic acid conjugate’s texture-stabilizing effect during storage. This could be attributed to the antioxidative and antibacterial activities of tannic acid in the whey protein–tannic acid conjugate in the sausage samples during storage. Tannic acid demonstrated antibacterial activity in striped catfish slices by reducing total viable count and psychrophilic bacterial count during refrigerated storage under 60% N_2_/35% CO_2_/5% O_2_ modified atmosphere packaging [18]. In terms of springiness (Table 1), there was no difference between control and the samples at any time point (*p* > 0.05), and the springiness of both sausages appeared to be reduced slightly, with values ranging from 8.7 to 8.2 mm. The results for cohesiveness ranged between 0.42–0.49 [Table 1]. At Day 0, the treated sample showed higher cohesiveness than the control, indicating that the whey protein–tannic conjugate had binding ability in the sausage matrix. Cohesiveness of the control remained constant with increasing storage time (*p* > 0.05), whereas it decreased slightly in the treated sample at Day 21 (*p* < 0.05). The changes in hardness, springiness, and cohesiveness may be related to protein–protein interactions, protein–water interactions, and protein–lipid interactions, all of which may be diminished as storage time increases.

It has also been noted that as storage time increases, the protein film enclosing fat globules in the emulsion system degrades [45]. The slowed lipid oxidation in samples containing WPC may mitigate the deleterious effects of oxidation products to some extent. Protein oxidation could be induced by radicals produced by lipid oxidation [46]. Lipid and protein oxidation are closely linked to the deterioration of meat products [47]. Protein oxidation has been shown to have a negative impact on the sensory quality of fresh meat and meat products in terms of texture, tenderness, and color [48]. A modest quantity of plant phenolic extract has been shown to delay protein oxidation in an emulsified pork sausage model system [27]. Overall, the results revealed that a whey protein–tannic acid conjugate could delay the softening of pork emulsion sausages that had been stored for an extended period of time.

The covalent crosslinking of tannic acid and whey protein at the aqueous phase was found to be important to the maintenance of the emulsion gel [7]. For example, gels with added 0.05% oxidized tannic acid had considerably lower expressible moisture content than the control [43]. In this study, the initial water-holding capacity of both sausages was not different (*p* > 0.05) (Table 1). However, as storage time increased, the water retention capacity of sausages with the whey protein–tannic acid conjugate remained stable for up to 7 days before decreasing marginally. The water-holding capacity tended to decline throughout storage for the control. On Day 7–21, sausages with the whey protein–tannic acid conjugate held more water than the control at all time points. This was most likely due to the whey protein–tannic acid conjugate’s increased emulsion stability index, which can maintain the balance between protein–protein and protein–water/lipid interactions to a greater extent than the control with free whey protein [20]. Furthermore, protecting muscle membranes from lipid oxidation with antioxidants helped maintain muscle fiber membrane integrity and minimize moisture loss, which had an effect on sausage textural qualities [49].

### 3.3. Microstructure

Figure 3 depicts the microstructures of pork emulsion sausage supplemented with whey protein (control) and the whey protein–tannic acid conjugate at Day 0 and Day 21 of refrigerated storage. Both pork emulsion sausages had a tight structure with fewer cavities on Day 0. This was consistent with both sausages having the same textural properties, particularly hardness, springiness, cohesiveness, and water-holding capacity (Table 1). Some tiny fat droplets can be seen in the control sausages (Figure 3a), but not in the sausages with the whey protein–tannic acid conjugate (Figure 3b). This was attributed to the fact that the emulsion stability index of the whey protein–tannic acid conjugate was higher than that of native whey protein [7,20].

According to Maqsood et al. [34], adding tannic acid at a concentration of 0.04% to fish emulsion sausage can contribute continuously and homogeneously dispersed oil droplets, conferring a greater consistency to the product and improving hardness. The microstructures of both sausages were altered after 21 days of cold storage to loosen the structure. The control sausage, on the other hand, displayed emulsion coalescence, as demonstrated by a rise in the number of oil droplets and large voids (Figure 3c). This was attributable to emulsion instability caused by protein network disorganization during refrigerated storage. Protein, in general, has emulsifying abilities and plays a key role in the stability of meat emulsions. Protein oxidation reduces protein solubility, causing aggregation and the creation of complex molecules via crosslinking. As a result, the protein’s emulsifying characteristics are reduced [50]. The use of the whey protein–tannic acid conjugate in pork emulsion sausage, on the other hand, can delay emulsion instability while also stabilizing the protein network (Figure 3d). According to Figure 3d, the microstructure of sausage containing the whey protein–tannic acid conjugate showed fewer oil droplets and voids. This finding concurred with that of Hayes et al. [44], who found that natural antioxidants such lutein, ellagic acid, and olive leaf extract could increase emulsion stability in cooked pork sausage by protecting proteins from oxidation. As a result, the whey protein–tannic acid conjugate can be used to retain the protein matrix and textural features of pork emulsion sausages during refrigerated storage, acting as antioxidants and preventing oxidative protein deterioration.

### 3.4. Color

Table 2 shows the color characteristics represented as *L** (lightness), *a** (redness), *b** (yellowness), and ΔE (total color difference) of pork emulsion sausages treated with whey protein–tannic acid in comparison to whey protein (control) during refrigerated storage for 21 days. When the whey protein–tannic acid conjugate was added, there was no difference in the *L**, *a**, and *b** values of the sausages when compared to the control (*p* > 0.05). However, there was a modest increase in ΔE value in samples containing the whey protein–tannic acid conjugate (*p* < 0.05). ΔE reflects the total color difference by taking into account all changes in the values of *L**, *a**, and *b**. It is feasible to forecast whether the consumer would detect a change in meat color based on ΔE [51]. According to Maqsood et al. [34], adding phenolic compounds, tannic acid, and ethanolic kiam wood extract (EKWE) to fish emulsion sausage may result in a drop in *L** and an increase in *a** due to the color of the phenolic compound employed. Using smaller concentrations of tannic acid and EKWE, however, had no effect on the color of the fish emulsion sausage. Due to the low concentration of tannic acid used in the conjugation, the colors of the whey protein–tannic acid conjugate and the natural whey protein were nearly identical in this study. As a result, the whey protein–tannic acid conjugate had no effect on the color of the final sausage. Both sausage samples appeared to alter color similarly during refrigerated storage for 21 days. Both sausage samples showed a considerable decrease in *L** and an increase in ΔE, while the redness *a** and *b** decreased slightly. There was no discernible difference in color between the two pork sausage samples after 7 days of storage. From Day 7 to Day 14, sausage containing the whey protein–tannic acid conjugate had a greater decrease in *L** and an increase in ΔE, whereas the control sausage showed only a slight change. The substantial change in *L** and ΔE was noticed in the control sausage after 21 days of storage. Changes in physical or chemical qualities of sausage, such as a loss of surface moisture, metmyoglobin production, lipid oxidation, and emulsion instability, are likely to cause discoloration. Some authors reported that using natural antioxidant extracts or phenolic compounds could stabilize the redness and lightness of meat products by slowing metmyoglobin formation and the establishment of discolored reactions, such as the Maillard browning reaction [27,52,53]. It was discovered in this study that whey protein–tannic acid had no negative effect on sausage production and appeared to aid in maintaining color stability.

## 4. Conclusions

When compared to sausage supplemented with ordinary whey protein, the addition of a whey protein–tannic acid conjugate to pork emulsified sausage could delay quality changes and help to extend product shelf-life. The antioxidant emulsifier effect of the whey protein–tannic acid conjugate may assist in maintaining oxidative stability and emulsion stability, textural qualities, and the color of the products. The TBARS of sausages containing the whey protein–tannic acid conjugate were lower than those of sausages containing regular whey protein after 21 days. According to the permitted requirement for TBARS level, the control sample can be stored for 7 days at 4–5 °C; however, the sample containing the whey protein–tannic acid conjugate can be stored for up to 21 days. Because the control and treated samples exhibited the same levels of cohesiveness, hardness, springiness, and water-holding capacity at Day 0, the addition of the whey protein–tannic acid conjugate had no impact on the product’s texture. The whey protein–tannic acid conjugate-added sausage did not display emulsion coalescence at Day 21, according to SEM results. When the whey protein–tannic acid conjugate was introduced, the *L**, *a**, and *b** values of the sausages remained unchanged. Thus, a whey protein–tannic acid conjugate might be thought of as a multifunctional ingredient for emulsion-type pork sausages. However, sensory assessment, a key quality element, can be expanded in the future to assure the efficacy of this strategy.

## Figures and Tables

**Figure 1 foods-12-02766-f001:**
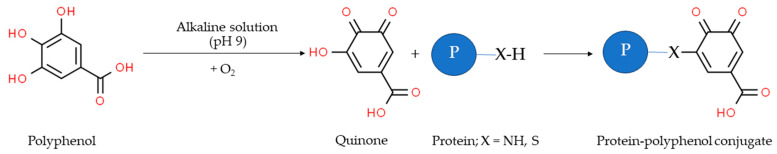
Proposed mechanism for covalent conjugation of protein and polyphenol.

**Figure 2 foods-12-02766-f002:**
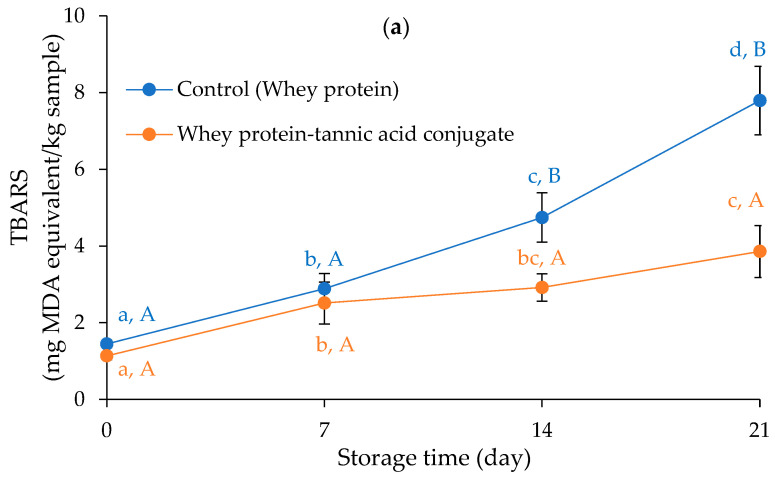
Changes in thiobarbituric acid reactive substances (TBARS) of pork emulsion sausages with added whey protein–tannic acid conjugate in comparison with control (whey protein) during refrigerated storage (**a**) and a kinetic analysis of the increase in lipid oxidation evaluated by TBARS of pork sausages during refrigerated storage (**b**). The first-order kinetic model was given as ln[TBARS_t_/TBARS_0_] = kt and the slope of the graph was used to calculate the reaction rate constant (k). MDA = malondialdehyde. Bars represent standard deviation from triplicate determinations. Different lowercase letters within the same sample indicate the significant difference (*p* < 0.05). Different uppercase letters within the same storage time indicate the significant difference (*p* < 0.05).

**Figure 3 foods-12-02766-f003:**
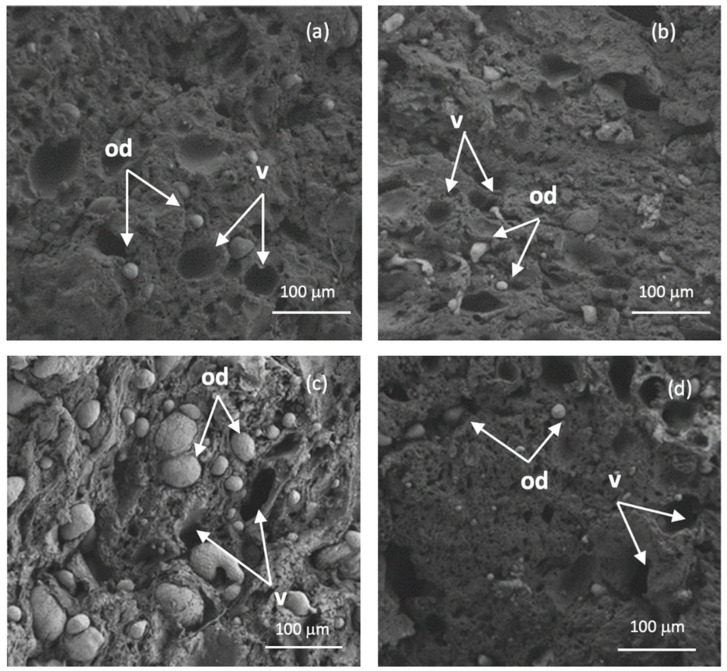
Changes in microstructure of pork emulsion sausages with added whey protein (control) (**a**,**c**) and whey protein–tannic acid conjugate (**b**,**d**) during refrigerated storage at Day 0 (**a**,**b**) and Day 21 (**c**,**d**) using 2 kV of acceleration and 200 times magnification. v = void. od = oil droplet.

**Table 1 foods-12-02766-t001:** Changes in hardness, springiness, cohesiveness, and water-holding capacity of pork emulsion sausages with added whey protein–tannic acid conjugate in comparison with control (whey protein) during refrigerated storage.

Samples	Days	Hardness (N)	Springiness (mm)	Cohesiveness	Water-Holding Capacity (%)
Control (Whey protein)	0	33.14 ± 1.26 ^bA^	8.52 ± 0.12 ^aA^	0.45 ± 0.02 ^aA^	93.01 ± 1.41 ^cA^
	7	24.54 ± 0.62 ^aA^	8.45 ± 0.17 ^aA^	0.47 ± 0.03 ^aA^	87.55 ± 1.90 ^bA^
	14	22.13 ± 2.20 ^aA^	8.47 ± 0.19 ^aA^	0.49 ± 0.03 ^aB^	84.24 ± 2.59 ^aA^
	21	22.55 ± 0.23 ^aA^	8.47 ± 0.33 ^aA^	0.43 ± 0.10 ^aA^	83.13 ± 5.92 ^aA^
Whey protein–tannic acid conjugate	0	33.31 ± 2.66 ^bA^	8.68 ± 0.04 ^bA^	0.48 ± 0.04 ^bA^	92.36 ± 1.02 ^aA^
	7	32.63 ± 4.84 ^abB^	8.66 ± 0.11 ^bA^	0.46 ± 0.03 ^abA^	92.44 ± 1.01 ^aB^
	14	25.89 ± 1.99 ^aB^	8.19 ± 0.14 ^aA^	0.42 ± 0.01 ^aA^	91.39 ± 1.82 ^aB^
	21	26.79 ± 2.99 ^aB^	8.20 ± 0.17 ^aA^	0.42 ± 0.01 ^aA^	90.84 ± 1.54 ^aB^

Values are mean ± standard deviation. Different lowercase letters within the same sample in the same column denote the significant difference (*p* < 0.05). Different uppercase letters within the same storage time in the same column denote the significant difference (*p* < 0.05).

**Table 2 foods-12-02766-t002:** Changes in *L**, *a**, *b**, and total color difference (ΔE) of pork emulsion sausages with added whey protein (control) and whey protein–tannic acid conjugate during refrigerated storage.

Samples	Days	*L**	*a**	*b**	ΔE
Control (Whey protein)	0	71.51 ± 0.14 ^bA^	2.25 ± 0.42 ^abA^	15.22 ± 0.32 ^bA^	0.00 ± 0.00 ^aA^
	7	72.28 ± 0.53 ^bA^	2.46 ± 0.05 ^bA^	15.39 ± 0.09 ^bB^	0.83 ± 0.51 ^bA^
	14	71.30 ± 0.53 ^bB^	2.17 ± 0.15 ^aA^	15.15 ± 0.11 ^bB^	0.46 ± 0.18 ^bA^
	21	64.30 ± 0.18 ^aA^	1.90 ± 0.49 ^aA^	13.65 ± 0.16 ^aA^	7.39 ± 0.51 ^cB^
Whey protein–tannic acid conjugate	0	72.74 ± 0.24 ^bA^	2.32 ± 0.08 ^bA^	15.28 ± 0.14 ^cA^	1.24 ± 0.25 ^bB^
	7	71.72 ± 0.87 ^bA^	2.41 ± 0.10 ^bA^	15.02 ± 0.09 ^bA^	0.79 ± 0.05 ^aA^
	14	67.30 ± 0.28 ^aA^	2.08 ± 0.10 ^aA^	14.24 ± 0.13 ^aA^	4.33 ± 0.25 ^cB^
	21	67.33 ± 1.02 ^aB^	2.00 ± 0.04 ^aA^	14.83 ± 0.21 ^bB^	4.20 ± 1.04 ^cA^

Values are mean ± standard deviation. Different lowercase letters within the same sample in the same column denote the significant difference (*p* < 0.05). Different uppercase letters within the same storage time in the same column denote the significant difference (*p* < 0.05).

## Data Availability

Data is contained within the article.

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
