# Peer review of "Whey Protein–Tannic Acid Conjugate Stabilized Emulsion-Type Pork Sausages: A Focus on Lipid Oxidation and Physicochemical Features"

_foods, 2023, doi:10.3390/foods12142766_

Round 1
Reviewer 1 Report
The authors present this manuscript related to Whey Protein-Tannic Acid Conjugate Stabilized Emulsion Type Pork Sausages. This manuscript is very specific to this topic; further authors try to mention only its lipid oxidation and physicochemical properties. The overall manuscript lacks novelty and new direction for the readers in this research area. Limited physicochemical properties investigated by authors. It seems just a report. I don’t find any results or discussion in-depth. Further, find my major and minor comments for improvement.
The manuscript contains unexpected plagiarism. Many sentences directly copied from previous publications.
At present abstract is so long seems the authors want to present in detail. Please shorten it.
Line 110, what are these words meaning DPPH, ABTS, and FRAP. Need to mention its full name.
Line 118 and 120 need to be mentioned in separate paragraphs about your work.
Line 71 and 72, the texts need to be elaborate. These two manuscripts can cite and helpful for authors about tannic acid; Materials Chemistry and Physics, 285, 126141 (2022); and Progress in Organic Coatings, 174, 107305 (2023).
Why authors choose pH 9 for the preparation of whey protein tannic acid conjugate? Any specific reason? Need to be mentioned in the manuscript.
Line 179, check the repeating of citations 35.
In my perspective, in the introduction section, you have mentioned about emulsion stability of processed meat products by adding additives and other ingredients. What are these additives you want to mention? I think you can write about how essential oil from different plant sources helps in combating antimicrobial activities in meat products. You can refer to this recent paper: https://doi.org/10.1016/j.tifs.2022.10.012
There is no interaction data for whey protein and tannic acid.
What is the mechanism of the interaction? Authors need to show in the form of a scheme.
For lipid oxidation, authors used Thiobarbituric Acid Reactive Substances (TBARS) assay. Authors need to other assay to show its lipid oxidation behaviour. One is not sufficient to prove its lipid oxidation behaviour.
Why authors taken specific conc. Of whey protein and tannic acid. Any specific reason. Did authors optimise it.
Line 215. need citations.
Kinetics results are not well agreed with theoretical results as its r2 is low below 0.9?
In figure 3, no scalar bar.
Colour data need to present in tabular format.
How authors did water capacity, which standard author follow. Same as hardness, sample how authors prepare.
Line 25: In place of Treatment write Treated
Line 51-52: Need to frame the sentence properly
Line 91: such “as”
Authors need to explain more about antioxidant activity is affected by combining whey protein and phenolic compounds?
In Section 2.4 Don’t write Analyses. Replace With physio-chemical properties.
Line 184: SEM magnification x should be capital “X”.
In some places Figure and Fig is mentioned in the text. Try to maintain the uniformity. Write it in one style. Also give space between Fig and “number”. Most preferably write it as “Fig. 1, Fig. 2” etc.
Fig. 3 caption is not in proper alignment.
Please check Fig. 4 the axes are not visible properly and Storage time (day) has been repeated. Also name a, b, c and d properly.
Please check English throughout the manuscript thoroughly.
English need to correct throughout the manuscript.
Author Response
Reviewer 1
Comments and Suggestions for Authors
The authors present this manuscript related to Whey Protein-Tannic Acid Conjugate Stabilized Emulsion Type Pork Sausages. This manuscript is very specific to this topic; further authors try to mention only its lipid oxidation and physicochemical properties. The overall manuscript lacks novelty and new direction for the readers in this research area. Limited physicochemical properties investigated by authors. It seems just a report. I don’t find any results or discussion in-depth. Further, find my major and minor comments for improvement.
The manuscript contains unexpected plagiarism. Many sentences directly copied from previous publications.
Ans: Thank you so much for your kind recommendation and for thoroughly reviewing our work. As previously stated, the focus of this work was on the lipid oxidation and physicochemical features of emulsion type pork sausages stabilized by whey protein-tannic acid conjugate. The research was carried on to discover the applicability of the whey protein-tannic acid conjugate, which has previously been studied in vitro for its antioxidant and interfacial properties. We attempted to calculate the rate of lipid oxidation rather than simply reporting the TBARS value in order to determine the influence of whey protein-tannic conjugate on lipid oxidation. The discussion was based on the findings, and the potential mechanism of action was also offered; this was not just a report. We were concerned about plagiarism as a means of preparing a quality manuscript. The original manuscript's similarity index was examined using Turnitin, and it also passed the editorial office's quality check using iThenticate. It implied that our manuscript had passed the similarity rate assessment.
At present abstract is so long seems the authors want to present in detail. Please shorten it.
Ans: The abstract was shorted as per suggestion.
Line 110, what are these words meaning DPPH, ABTS, and FRAP. Need to mention its full name.
Ans: It was changed to “In a prior work [20], whey protein coupled with 5% oxidized tannic acid displayed the highest antioxidative activity, as determined by 1,1-diphenyl-2- picrylhydrazyl (DPPH), 2,2-azinobiz-ethylbenzothiozolie-6-sulphonic acid (ABTS), and ferric reducing antioxidant power (FRAP) assays, along with an enhanced emulsion stability index in vitro.”
Line 118 and 120 need to be mentioned in separate paragraphs about your work.
Ans: Done.
Line 71 and 72, the texts need to be elaborate. These two manuscripts can cite and helpful for authors about tannic acid; Materials Chemistry and Physics, 285, 126141 (2022); and Progress in Organic Coatings, 174, 107305 (2023).
Ans: It was changed to “Tannic acid is a polyphenol with favorable physiological effects (such as anti-oxidant and antibacterial properties), and it can be utilized to create composite films as sustainable antioxidant and barrier packaging materials. Tannic acid can be modified to improve its antibacterial activity and biocompatibility for biomedical purposes [14, 15].” The references were updated.
Kim, H., Panda, P. K., Sadeghi, K., & Seo, J. (2023). Poly (vinyl alcohol)/hydrothermally treated tannic acid composite films as sustainable antioxidant and barrier packaging materials. Progress in Organic Coatings, 174, 107305.
Kim, H., Panda, P. K., Sadeghi, K., Lee, S., Chung, C., Park, Y., ... & Seo, J. (2022). Facile thermal and hydrolytic conversion of tannic acid: Enhancement of antimicrobial activity and biocompatibility for biomedical applications. Materials Chemistry and Physics, 285, 126141.
Why authors choose pH 9 for the preparation of whey protein tannic acid conjugate? Any specific reason? Need to be mentioned in the manuscript.
Ans: It was added in the section 2.2 Preparation of Whey Protein-Tannic Acid Conjugate that “Tannic acid is required to oxidize to quinone at a high pH range, pH 9 in this case, to allow the covalent interaction between tannic acid and whey protein.”
Also, it was originally stated in the second paragraph of the Introduction that “Covalent bonds do not easily break under some extreme conditions, such as heating, acid or alkali conditions, and so on [9, 10], making covalent-binding interaction or molecular modification a good solution to this issue. Molecular modification is an intriguing technique for creating novel materials with desired chemical and physical characteristics [11]. Because of polyphenols' numerous functions, protein-polyphenol conjugations have garnered a lot of interest. It has been observed that polyphenols can react with proteins under oxidizing circumstances, resulting in the creation of protein-polyphenol conjugates [10].”
Line 179, check the repeating of citations 35.
Ans: Done.
In my perspective, in the introduction section, you have mentioned about emulsion stability of processed meat products by adding additives and other ingredients. What are these additives you want to mention? I think you can write about how essential oil from different plant sources helps in combating antimicrobial activities in meat products. You can refer to this recent paper: https://doi.org/10.1016/j.tifs.2022.10.012
Ans: Because we would emphasize whey protein in the next section to discuss our study, we initially mentioned protein additives to stabilize emulsion in meat products. “Soy, casein, whey, egg white, gelatin, protein hydrolysates, and blood plasma are the proteins most frequently found in protein preparations used as emulsifiers and stabilizers in the creation of emulsion-type sausages [4, 5]. The key factors influencing their extensive use are their significant capacity to emulsify fat, stabilize the emulsion, and have substantial water-binding properties [4]. These proteins can be employed to lower production and product costs while improving the product's texture and cooking efficiency. Due to their nutritional and healthful properties or a lower energy value than animal fat, vegetable or whey preparations utilized in higher quantities become good replacements for meat and fat components [4].”
Thus, the essential oil is outside of our purview. We left out additional information regarding essential oils.
There is no interaction data for whey protein and tannic acid.
Ans: It was originally stated in the Introduction that “Protein -NH2 and -SH residues were able to form covalent C-N or C-S bonds with the C=O or C-H groups of tannic acid under alkaline circumstances, resulting in protein-tannic acid conjugates [19]. It was discovered that the conjugates stabilized emulsion was more stable against heat treatment and centrifugation compared to unmodified whey protein [7]. Covalent crosslinking in whey protein-tannic acid conjugates generated a network based on tannic acid molecules acting as bridges between emulsion droplets, which was significant in the stabilizing impact [7, 19]. Covalent connections were formed between whey protein and tannic acid, according to the results. Whey protein-tannic acid conjugates outperformed free whey protein in terms of oil-water interfacial activity and wettability [7, 20].”
What is the mechanism of the interaction? Authors need to show in the form of a scheme.
Ans: It was originally stated in the Introduction that “Protein -NH2 and -SH residues were able to form covalent C-N or C-S bonds with the C=O or C-H groups of tannic acid under alkaline circumstances, resulting in protein-tannic acid conjugates [19].”
For lipid oxidation, authors used Thiobarbituric Acid Reactive Substances (TBARS) assay. Authors need to other assay to show its lipid oxidation behaviour. One is not sufficient to prove its lipid oxidation behaviour.
Ans: Due to the fact that TBARS is a secondary lipid oxidation product and is connected to the rancidity and acceptability of sausage products. As a result, the TBARS analysis was employed in this study as one of the most crucial aspects of sausage quality.
Why authors taken specific conc. Of whey protein and tannic acid. Any specific reason. Did authors optimise it.
Ans: It was reported in our previous article and we highlighted this in the Introduction. “In a prior work [20], whey protein coupled with 5% oxidized tannic acid displayed the highest antioxidative activity, as determined by 1,1-diphenyl-2-picrylhydrazyl (DPPH), 2,2-azinobiz-ethylbenzothiozolie-6-sulphonic acid (ABTS), and ferric reducing antioxidant power (FRAP) assays, along with an enhanced emulsion stability index in vitro.The linkage between whey protein and phenolic compounds was validated by a loss in free amino content and an increase in total phenolic content of the modified whey protein after it was treated with oxidized phenolic compounds [20]. Tannic acid with a high hydroxyl and carboxyl group concentration was incorporated into whey protein, which increased its hydrophilicity. When the hydrophilic group increased faster than the hydrophobic group, the surface hydrophobicity of the whey protein treated with oxidized tannic acid decreased [20]. It was proposed that whey protein treated with 5% oxidized tannic acid might be employed as an antioxidative food additive in emulsified food products.”
Line 215. need citations.
Ans: Done.
Kinetics results are not well agreed with theoretical results as its r2 is low below 0.9?
In figure 3, no scalar bar.
Ans: The R2 refers to how well the linearity of the oxidation progression rate is. Both treatments exhibit a linear trend, with the control having a higher R2 (0.9666) due to spontaneous oxidation in the absence of an antioxidant. However, the treatment with whey protein-tannic acid conjugate exhibited a lower R2 (0.8686), which could be attributed to delayed lipid oxidation during some storage periods due to the conjugate's antioxidant action. As a result, the rate of lipid oxidation did not increase dramatically.
Also, the SD bars were added and the letters indicating significant differences were incorporated.
Colour data need to present in tabular format.
Ans: Done.
How authors did water capacity, which standard author follow. Same as hardness, sample how authors prepare.
Ans: The procedures for determining texture and water holding capacity were given more details.
Line 25: In place of Treatment write Treated
Ans: Done.
Line 51-52: Need to frame the sentence properly
Ans: It was changed to “The key factors influencing their extensive use are their significant capacity to emulsify fat, stabilize the emulsion, and have substantial water-binding properties [4].”
Line 91: such “as”.
Ans: Done.
Authors need to explain more about antioxidant activity is affected by combining whey protein and phenolic compounds?
Ans: Some of this information was covered in our earlier research, which was cited in this paper. “Furthermore, the protein-phenolic conjugate has been shown to be a natural antioxidant for meat products [31]. Whey protein is a typical food addition used to improve product stability and texture, whereas phenolics are food antioxidants [20]. As a result, coupling whey protein with phenolics is an efficient strategy to improve protein functioning while preserving polyphenol bioactivity [20]. In a prior work [20], whey protein coupled with 5% oxidized tannic acid displayed the highest antioxidative activity, as determined by 1,1-diphenyl-2- picrylhydrazyl (DPPH), 2,2-azinobiz-ethylbenzothiozolie-6-sulphonic acid (ABTS), and ferric reducing antioxidant power (FRAP) assays, along with an enhanced emulsion stability index in vitro.”
In Section 2.4 Don’t write Analyses. Replace With physio-chemical properties.
Ans: Done.
Line 184: SEM magnification x should be capital “X”.
Ans: Done.
In some places Figure and Fig is mentioned in the text. Try to maintain the uniformity. Write it in one style. Also give space between Fig and “number”. Most preferably write it as “Fig. 1, Fig. 2” etc.
Ans: In the text, it was written as "(Fig. x)," yet at the start of the sentence, it was written as "Figure x......."
Fig. 3 caption is not in proper alignment.
Ans: The caption was enhanced, and Figure 3 was changed to Figure 2. In the figures, editing tools were used to highlight the voids and fat droplets.
Please check Fig. 4 the axes are not visible properly and Storage time (day) has been repeated. Also name a, b, c and d properly.
Ans: Figure 4 was changed to Table to make it simpler to understand, and the letters indicating significant differences were added.
Please check English throughout the manuscript thoroughly.
Ans: QuillBot, a paraphrase program, was used to review the English.

Reviewer 2 Report
The manuscript foods-2468900 entitled "Whey Protein-Tannic Acid Conjugate Stabilized Emulsion-Type Pork Sausages: A Focus on Lipid Oxidation and Physicochemical Features”
Overall, the paper is well written and very well explained.
In its current state, the level of English throughout your manuscript does not meet the journal's desired standard. There are many badly worded/constructed sentences.
Abstract
The main content of the abstract should include the briefly purpose of the research, the principal result and major conclusion. The abstract, in the present form is very disorganized and too long. Please revise it to maximum 150-200 words.
Line 11-13: Please delete the following sentences, not required here “Emulsion-type pork sausages are a meat product made primarily of pork and a variable proportion of fat that is chopped and blended with non-meat substances. The quality of the products may deteriorate during storage due to lipid oxidation and emulsion instability”.
Keywords: Please replace the words which are already in the main title such as whey; tannic acid; conjugate; lipid oxidation; sausages.
Introduction
The introduction in the present form is very disorganized. For instance, first talk about the lipid oxidation issue then come towards solution like using tannic acid. Please rearrange it.
Materials and Methods
Line 132: Why pH was adjusted to 9? Is it safe to consume a product with a pH level around 9?
2.4.1. Thiobarbituric Acid Reactive Substances (TBARS)
Please mention the days 0, 7, 14, and 21 here for recording of TBARS values.
Please delete the following sentences. No need to explain the already well-known techniques. Proper citation is enough
“Samples 160 (0.5 g) were combined with 2.5 mL of a TBA solution containing 0.375% TBA, 15% TCA and 0.25 N HCl. The combinations were heated in boiling water for 10 min, cooled with running tap water, and then centrifuged at 5000 ´g for 10 min at 25 °C. The absorbance of the supernatant was measured at 532 nm. A standard curve was created using 1,1,3,3 tetramethoxypropane at concentrations ranging from 0 to 10 ppm. TBARS were represented as mg of malondialdehyde (MDA) equivalent/kg sample”.
2.4.3. Microstructure
Please explain the sample processing method. Is it freeze dried? If yes, please explain
Results and discussion
Figure 3
Why Figure C is totally different from other three?
Please mark the fat droplets in the figures using editing tools.
I am not satisfied with this figure.
Please convert Texture profile analysis and color parameters graphs to Table showing level of significance and standard deviation.
In its current state, the level of English throughout your manuscript does not meet the journal's desired standard. There are many badly worded/constructed sentences.
Author Response
Reviewer 2
Comments and Suggestions for Authors
The manuscript foods-2468900 entitled "Whey Protein-Tannic Acid Conjugate Stabilized Emulsion-Type Pork Sausages: A Focus on Lipid Oxidation and Physicochemical Features”
Overall, the paper is well written and very well explained.
Ans: Thank you very much.
In its current state, the level of English throughout your manuscript does not meet the journal's desired standard. There are many badly worded/constructed sentences.
Ans: QuillBot, a paraphrase program, was used to refine the English.
Abstract
The main content of the abstract should include the briefly purpose of the research, the principal result and major conclusion. The abstract, in the present form is very disorganized and too long. Please revise it to maximum 150-200 words.
Ans: Abstract was revised accordingly.
Line 11-13: Please delete the following sentences, not required here “Emulsion-type pork sausages are a meat product made primarily of pork and a variable proportion of fat that is chopped and blended with non-meat substances. The quality of the products may deteriorate during storage due to lipid oxidation and emulsion instability”.
Ans: Done.
Keywords: Please replace the words which are already in the main title such as whey; tannic acid; conjugate; lipid oxidation; sausages.
Ans: Keywords were changed to “phenolic-protein conjugate; protein modification; functional properties; meat; stability”
Introduction
The introduction in the present form is very disorganized. For instance, first talk about the lipid oxidation issue then come towards solution like using tannic acid. Please rearrange it.
Ans: Because there are two issues with emulsion type sausages: emulsion instability and lipid oxidation. So, in the first paragraph, general information about emulsion-type sausages was provided, as well as the use of protein preparations as emulsifiers and stabilizers in the creation of emulsion-type sausages, of which whey protein is one of the candidates.
The details and limitations of whey protein were then presented, which led to the modification of whey protein to improve its emulsifying properties. Following that, whey protein-phenolic conjugate was mentioned as a successful modification, with an emphasis on tannic acid.
Then, the second concern, lipid oxidation, was mentioned, which can be postponed utilizing whey protein-tannic acid conjugate, as observed in our prior in vitro work. This leads to the objective of this study as an application element in the last section of the Introduction.
Materials and Methods
Line 132: Why pH was adjusted to 9? Is it safe to consume a product with a pH level around 9?
Ans: It was added in the section 2.2 Preparation of Whey Protein-Tannic Acid Conjugate that “Tannic acid is required to oxidize to quinone at a high pH range, pH 9 in this case, to allow the covalent interaction between tannic acid and whey protein.”
Also, it was originally stated in the second paragraph of the Introduction that “Covalent bonds do not easily break under some extreme conditions, such as heating, acid or alkali conditions, and so on [9, 10], making covalent-binding interaction or molecular modification a good solution to this issue. Molecular modification is an intriguing technique for creating novel materials with desired chemical and physical characteristics [11]. Because of polyphenols' numerous functions, protein-polyphenol conjugations have garnered a lot of interest. It has been observed that polyphenols can react with proteins under oxidizing circumstances, resulting in the creation of protein-polyphenol conjugates [10].”
The product is safe to consume since the pH of modified whey protein is neutralized following dialysis against distilled water. It was stated in the text that “The pH of the conjugate after dialysis was around 7.0.”
2.4.1. Thiobarbituric Acid Reactive Substances (TBARS)
Please mention the days 0, 7, 14, and 21 here for recording of TBARS values.
Ans: It was changed to “TBARS were investigated at the Days 0, 7, 14, and 21 using the method outlined by Buege and Aust [32].”
Please delete the following sentences. No need to explain the already well-known techniques. Proper citation is enough “Samples 160 (0.5 g) were combined with 2.5 mL of a TBA solution containing 0.375% TBA, 15% TCA and 0.25 N HCl. The combinations were heated in boiling water for 10 min, cooled with running tap water, and then centrifuged at 5000 ´g for 10 min at 25 °C. The absorbance of the supernatant was measured at 532 nm. A standard curve was created using 1,1,3,3 tetramethoxypropane at concentrations ranging from 0 to 10 ppm. TBARS were represented as mg of malondialdehyde (MDA) equivalent/kg sample”.
Ans: Done.
2.4.3. Microstructure
Please explain the sample processing method. Is it freeze dried? If yes, please explain

Reviewer 3 Report
The research topic is certainly of interest. However, it would be beneficial to make several corrections and additions to improve the overall quality of the paper. Here are my detailed suggestions:
- The abstract seems lengthy. Consider reducing it to succinctly capture the main points of your study.
- On line 120, the objective in the introduction could be better articulated. Also, it's not clear how the control was formulated with whey protein.
- Line 145: Could you clarify the intent of the statement, “The moisture of the mince was set at 86%”?
- Lines 146 – 147: It would be helpful if you could provide a reason for using a phosphate quantity three times greater than what is permissible in many countries.
- Line 148: Could you elaborate on why you chose to use soybean oil instead of pork backfat?
- Lines 160 – 162: The TBARS analysis explanation needs improvement. It's not clear how malonaldehyde extraction was performed based on your current description.
- Lines 172 – 174: Could you provide more detail on the texture analysis? Why were only hardness and springiness evaluated?
- Lines 175 – 180: The technique you've used here might not be suitable for this type of product. Why didn't you analyze emulsion stability?
- Lines 180 – 184: This determination could use more detail.
- Lines 189 – 191: Could you clarify how Delta E was calculated? Was it computed within the same treatment comparing days 7, 14, and 21 with day 0?
- Line 194: Could you explain how the analyses were replicated?
- Please include letters indicating statistical differences in the figures.
- Line 200: The TBARS values seem quite high. It would be useful to compare these with literature values.
- Lines 211 – 212: The value you've mentioned here seems incorrect. Other authors assert that for meat products, values greater than 1 mg MDA/kg sample indicate that the product is oxidized.
- Figure 2: Please present the results for cohesiveness.
- Why wasn't pH evaluated?
- Line 391: Emulsion stability wasn't analyzed, could you clarify why?
I believe addressing these points would greatly enhance the clarity and overall quality of your research article.
The quality of the English language used in the article is generally acceptable. Most of the content is understandable and the scientific terminology is used correctly. However, some sentences may benefit from revision for clarity and coherence. Addressing the aforementioned points will not only improve the scientific quality of your research article, but will also enhance its readability.
Author Response
Reviewer 3
Comments and Suggestions for Authors
The research topic is certainly of interest. However, it would be beneficial to make several corrections and additions to improve the overall quality of the paper. Here are my detailed suggestions:
- The abstract seems lengthy. Consider reducing it to succinctly capture the main points of your study.
Ans: Abstract was revised accordingly.
- On line 120, the objective in the introduction could be better articulated. Also, it's not clear how the control was formulated with whey protein.
Ans: It was changed to “As a result, the purpose of this study was to investigate the impact of whey protein-tannic acid conjugate on the oxidative stability and physicochemical features of pork emulsion sausages in comparison with unmodified whey protein.”
- Line 145: Could you clarify the intent of the statement, “The moisture of the mince was set at 86%”?
Ans: The moisture content of the pork is checked before it is used. It was stated that “To maintain consistency between lots, the moisture content of the mince was set at 86%.”
- Lines 146 – 147: It would be helpful if you could provide a reason for using a phosphate quantity three times greater than what is permissible in many countries.
Ans: I appreciate your feedback. We erred in doing so. The actual amount of sodium tripolyphophate utilized was 0.25 g. Rechecking and fixing the formula were done.
- Line 148: Could you elaborate on why you chose to use soybean oil instead of pork backfat?
Ans: The statement was added. “In contrast to pork backfat, which mostly contains saturated fat, soybean oil was intended to use in this study since it can be simpler to oxidize and generate coalescence. With this oil, it may be easier for the system to keep track of emulsion instability and lipid oxidation.”
- Lines 160 – 162: The TBARS analysis explanation needs improvement. It's not clear how malonaldehyde extraction was performed based on your current description.
Ans: Due to the suggestion of Reviewer2, the detail of TBARS analysis was deleted. Reviewer 2 said that “Please delete the following sentences. No need to explain the already well-known techniques. Proper citation is enough.”
However, the determination of TBARS using the Buege and Aust technique [32] is based on the development of a pink color brought on by the interaction of thiobarbituric acid (TBA) with malonaldehyde (MDA). MDA was extracted under acidic conditions, and upon heating, it further interacted with TBA to produce a pink adduct. It is the spectrophotometric determination of pink fluorescent MDA-TBA complex produced after reaction with TBA at low pH and high temperature.
- Lines 172 – 174: Could you provide more detail on the texture analysis? Why were only hardness and springiness evaluated?
Ans: The hardness, springiness, and cohesiveness were reported in this study. The detail on the texture analysis was given. “Hardness (N), springiness (mm), and cohesiveness of samples were measured using the methods described by Maqsood et al. [34], using a texture analyser (LR 5K; LLOYD instruments, West Sussex, England). Before analysis, the saples were left at room temperature for 1 h to equilibrate before being cut into cylinders (2 ´ 2 cm2). With a cylindrical aluminum probe with a diameter of 70 mm, two cycles of compression were carried out at a test speed of 1.0 mm/s (or 50% of the initial gel height) and trigger force of 0.05 N.”
- Lines 175 – 180: The technique you've used here might not be suitable for this type of product. Why didn't you analyze emulsion stability?
Ans: Since the sausage is an emulsion gel product, it is appropriate to evaluate its water holding capacity by measuring the expressible drip from the matrix. The sausage, which is an emulsion gel, may not be a good fit for the emulsion stability. However, the lipid coalescence in the SEM images can be used as an indirect technique for determining the emulsion stability, thus we did monitor the microstructural changes.
- Lines 180 – 184: This determination could use more detail.
Ans: The detail was given. “A scanning electron microscope (SEM) (GeminiSEM; Carl Ziess Microscopy, Germany) was used to examine the microstructures of cooked pork emulsion sausages [36]. The samples were cut into 1-2 mm cubes, fixed with 2.5% (v/v) glutaraldehyde in 0.2 M phosphate buffer (pH 7.2), and then rinsed twice with 0.1 M phosphate buffer and distilled water for 2 h at room temperature. After that, ethanol was used to dehydrate the fixed samples for 10 min at various concentrations of 50%, 70%, 80%, 90%, and 100%. After critical point drying with CO2 as the transition fluid, dried samples were mounted on a bronze stub and given a gold sputter coating. The samples were scanned using an SEM with a 3 kV acceleration voltage and a magnification of 200X.”
- Lines 189 – 191: Could you clarify how Delta E was calculated? Was it computed within the same treatment comparing days 7, 14, and 21 with day 0?
Ans: It was stated that “Total difference in color (DE) between the control sausage sample at Day 0 and that of other samples at Days 0, 7, 14, and 21 were calculated [35].” It means that the L*, a*, and b* of each sample were compared to the L*, a*, and b* of the control Day 0 to calculate the delta E of all samples. Only the delta E of control Day 0 is therefore zero.
- Line 194: Could you explain how the analyses were replicated?
Ans: It was stated that “All experiments were carried out in triplicate (n = 3) and the data were reported by mean ± standard deviation.”
- Please include letters indicating statistical differences in the figures.
Ans: Done.
- Line 200: The TBARS values seem quite high. It would be useful to compare these with literature values.
Ans: A comparison was made using the report of Laurouche et al. [37].
- Lines 211 – 212: The value you've mentioned here seems incorrect. Other authors assert that for meat products, values greater than 1 mg MDA/kg sample indicate that the product is oxidized.
Ans: Thank you very much. Legislative limits on MDA levels in meat samples do not exist, but some research have suggested that levels exceeding 1.0 mg/kg may be inappropriate. However, some studies have a varying range depending on the samples. Here, the classification of oxidized lipid based on TBARS was taken from Laurouche et al. [37]. So, further information about TBARS and oxidized lipid is provided. “In general, lipids can be classified as not oxidized (TBARS value < 1.5 mg MDA/kg), moderately oxidized (1.6 < TBARS value < 3.6), or oxidized (TBARS value > 3.7), according to Larouche et al. [37]. According to the permissible requirement for TBARS level, the control sample can be stored for 7 days, however the sample containing whey protein-tannic acid conjugate can be stored for up to 21 days.” Additionally, when we smelled the samples ourselves, we found no rancid odor.
- Figure 2: Please present the results for cohesiveness.
Ans: Cohesiveness was added and the Figure was change to Table.
- Why wasn't pH evaluated?
Ans: The pH values of all samples ranged from 6.7 to 6.8, and there was no significant difference over the course of storage. This information was added to the section 2.3 Preparation of Pork Emulsion Sausages.
- Line 391: Emulsion stability wasn't analyzed, could you clarify why?
Ans: The sausage, which is an emulsion gel, may not be a good fit for the emulsion stability. However, the lipid coalescence in the SEM images can be used as an indirect technique for determining the emulsion stability, thus we did monitor the microstructural changes.
I believe addressing these points would greatly enhance the clarity and overall quality of your research article.
Ans: Thank you very much for your invaluable suggestion.
Comments on the Quality of English Language
The quality of the English language used in the article is generally acceptable. Most of the content is understandable and the scientific terminology is used correctly. However, some sentences may benefit from revision for clarity and coherence. Addressing the aforementioned points will not only improve the scientific quality of your research article, but will also enhance its readability.
Ans: Thank you very much. QuillBot, a paraphrase program, was used to review the English.

Round 2
Reviewer 1 Report
The authors improved the manuscript. However, major questions were not solved. Therefore, this manuscript cannot be considered further. The following question needs to solve.
1. Last paragraph of the introduction needs to be the strength of your present study. At present, it is very simple.
2. “There is no interaction data for whey protein and tannic acid”. This question is not addressed. Of course, a similar study was done before. You prepare this sample in this study, so you must provide interaction experiment data.
3. What is the mechanism of the interaction? Authors need to show in the form of a scheme. In this question, you must show the interaction in the scheme.
4. In conclusion, insert some major obtained data, including the main key points.
Moderate editing of the English language required
Reviewer 3 Report
No further comments.
Author Response
Thank you very much for your invaluable suggestion.